# Mapping the Species Richness of Woody Plants in Republic of Korea

Junhee Lee [1], Youngjae Yoo [1], Raeik Jang [2] and Seongwoo Jeon [1]

1 Department of Environmental Science & Ecological Engineering, Korea University, Seoul 02841, Republic of Korea

2 Ojeong Resilience Institute, Korea University, Seoul 02841, Republic of Korea

* Correspondence: eepps_korea@korea.ac.kr; Tel.: +82-2-3290-3543

**Abstract:** As climate change continues to impact the planet, the importance of forests is becoming increasingly emphasized. The International Co-operative Program on the Assessment and Monitoring of Air Pollution Effects on Forests (ICP Forests) has been monitoring and assessing forests in 40 countries since 1985. In Republic of Korea, the first Forest Health Management (FHM) survey was a nationwide sample point assessment conducted between 2011 and 2015. However, there are limitations in representing the health of forests that occupy 63.7% of Korea's land area due to the nature of sample point surveys, which survey a relatively small area. Accordingly, a species richness map was created to promote species diversity in forest health evaluations in Republic of Korea. The map was created using data from the first FHM survey, which examined 28 factors with 12 survey indicators in four categories: tree health, vegetation health, soil health, and atmospheric health. We conducted an ensemble modeling of species distribution for woody plant species that are major habitats in Republic of Korea. To select the species, we used the first FHM survey data and chose those with more than 100 sample points, resulting in a total of 11 species. We then created the species richness map of Republic of Korea by overlaying their distributions. To verify the accuracy of the derived map, an independent verification was conducted using statistical verification and external data from the National Natural Environment Survey. To support forest management that accounts for climate change adaptation, the derived species richness map was validated based on the vegetation climate distribution map of the Korean Peninsula, which was published by the Korea National Arboretum. The map confirmed that species richness is highest around the boundary of the deciduous forest in the central temperate zone and lowest around the evergreen and deciduous mixed forest in the southern temperate zone. By establishing this map, it was possible to confirm the spatial distribution of species by addressing the limitations of direct surveys, which are unable to represent all forests. However, it is important to note that not all factors of the first FHM survey were considered during the spatialization process, and the target area only includes Republic of Korea. Thus, further research is necessary to expand the target area and include additional items.

**Keywords:** forest health management; species diversity; species distribution model; multi-model ensemble

## 1. Introduction

Forests offer numerous benefits to humans, including recreation, air purification, and water conservation [1]. However, the health of forest ecosystems is increasingly threatened by factors such as global warming caused by climate change, and efforts are being made to identify and manage changes in these ecosystems [2]. The International Co-operative Program on Assessment and Monitoring of Air Pollution Effects on Forests (ICP Forests) has evaluating and monitoring forest health in 40 countries since 1985. The Montreal Process [3] has established seven standards for systematic forest ecosystem management and health identification, with the third criterion, which focuses on "Maintaining the health and vitality of the forest ecosystem", being applied to establish survey indicators and conducts

forest surveys to evaluate forest health [4]. The Food and Agriculture Organization (FAO) has defined forest health by examining changes in forests based on theories that combine abiotic and biotic stressors. In addition, assessments of the yield and quality of timber and non-timber forest products and forest health have been conducted, focusing on recreational, scenic, and cultural values [5]. Furthermore, they defined healthy forests as those that exhibit resilience, and evaluated forest health based on these conditions [6].

On a global level, research is being conducted to understand the health of forests, and the collection of the current status of forests and basic data is necessary for systematic management of forest ecosystems. Domestically and internationally, research is being conducted through monitoring surveys of the forest ecosystem to evaluate its health. In Republic of Korea, the Korea Forest Service conducted the 5th National Inventory (2006–2010) with 4000 plots as fixed sample points based on the 5th Basic Forest Plan (2008–2017), of which 967 sample points were used. The first Forest Health Management (FHM) survey and analysis consisted of a total of 28 items with 12 survey indicators in 4 categories: tree health, vegetation health, soil health, and atmospheric health. The survey concluded that about 81.2% of the forests were healthy (National Institute of Forest Science, 2015). Overseas, between June and September every year, around 8400 plants across the UK are investigated in 350 monitoring plots to evaluate the health of five major tree species inhabiting the region (Norway spruce, Sitka spruce, spruce, Scots pine, oak, and beech); this is conducted using a tree-oriented survey data approach [7]. Similarly, in the USA, forest health was evaluated by dividing a 764-acre area into 10 natural populations in New Hampshire [8].

Most of the methods used in the field of forest environment are monitoring survey methods, which can be challenging to use when aiming to represent the environment as a continuous space [9]. For example, the first FHM surveys, as well as domestic and foreign forest ecosystem surveys, used sample point monitoring survey methods that are centered on stand surveys at specific sample points. However, this method represents only the health of the stand and is subject to investigator bias, making it less objective.

A forest is a complex assemblage of various species that coexist in a state of harmony with each other. Given that the concept of a forest is developed based on vegetation, species richness assumes a critical role in the description and analysis of forest health. In this study, we deemed it advantageous to express species richness using data from the first FHM survey. We conducted a study aimed at mapping the species richness of national forests featuring complex terrain characteristics and various species by establishing a connection between species richness and forest management policy, we addressed an important research topic.

## 2. Materials and Methods

### 2.1. Study Areas

This study focuses on Republic of Korea, which experiences four distinct seasons and is located between 33° N and 38° N and between 126° E and 132° E. The central region has a cold and wet climate in winter and a cold humid climate otherwise, whereas the southern region has a temperate rain climate and a warm and humid climate otherwise (Figure 1). Forests in Republic of Korea mainly consist of coniferous, deciduous, and mixed forests, which account for approximately 37%, 32%, and 26% of the total forest area, respectively, as of 2018. The FAO selected Republic of Korea as the only country that achieved successful forest reclamation and effective economic growth after World War II, making it an ideal subject for this study.

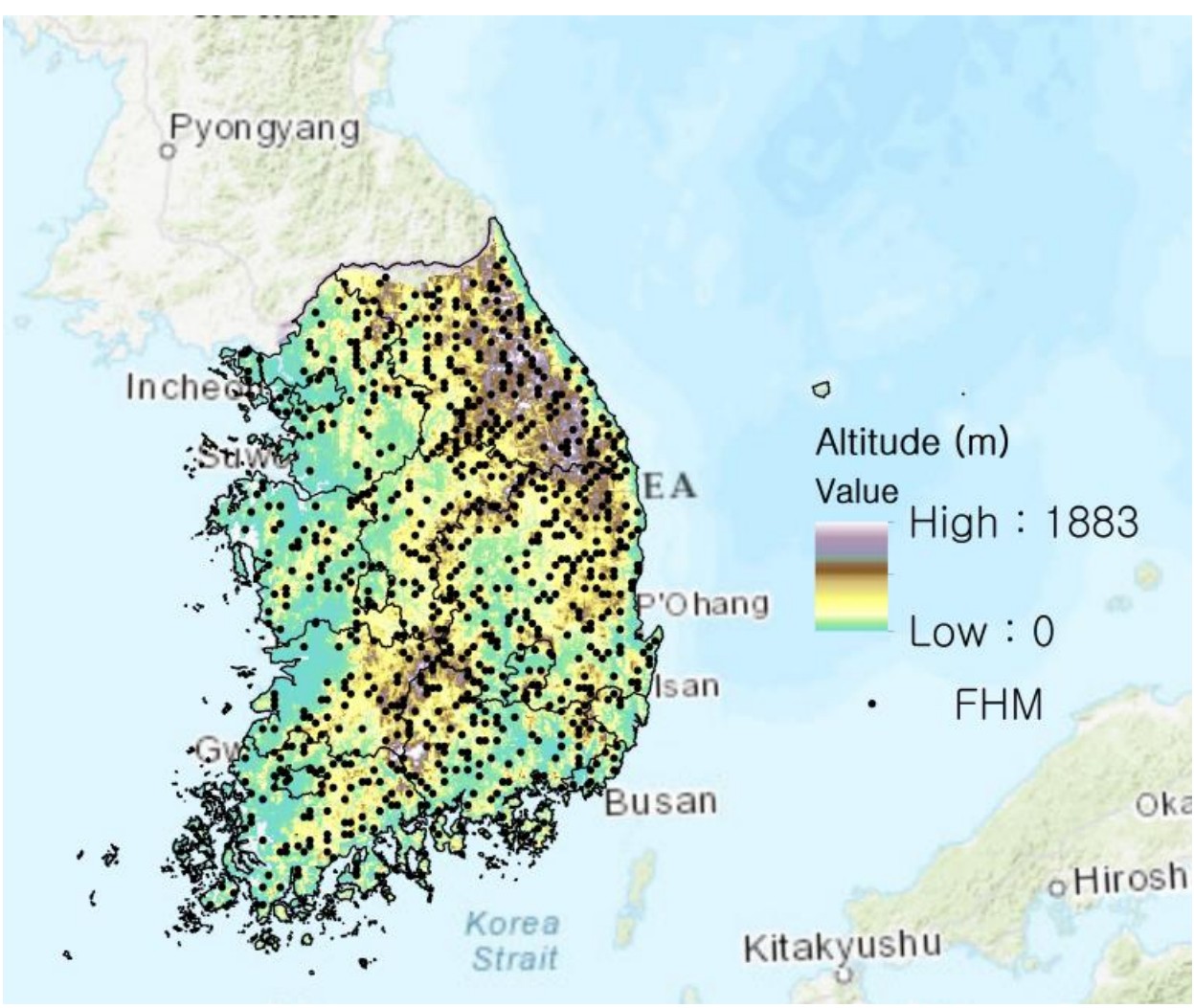

**Figure 1.** Study area and the First Forest Health Management Survey Points.

*2.2. First Forest Health Management Survey and Target Species*

The first FHM survey consisted of 28 items and 12 survey indicators in four categories: tree health, vegetation health, soil health, and atmospheric health. The first FHM survey was conducted by the Korea Forest Service based on the fifth National Inventory (2006–2010). For the fifth Basic Forest Plan (2008–2017), 4000 plots of fixed sample points were surveyed, with about 1/4 of them sampled, resulting in 967 sample points and 44,440 surveyed trees. The health and vitality of forests was also investigated [10]. Among the 28 items investigated, the National Institute of Forest Science selected 7 important indicators for forest health. In this study, the species diversity index, which can best express vegetation health among the seven indicators, was calculated based on species composition data [10] (Figure 2).

We selected the most abundant species in Korea as the target species by referring to a previous study [7] that had evaluated forest health for the most abundant species in the UK (Table 1). The selection process for the target species in the current study involved choosing those that were present in the first FHM basic survey zone and trees with large trunks in the survey zone (Figure 2).

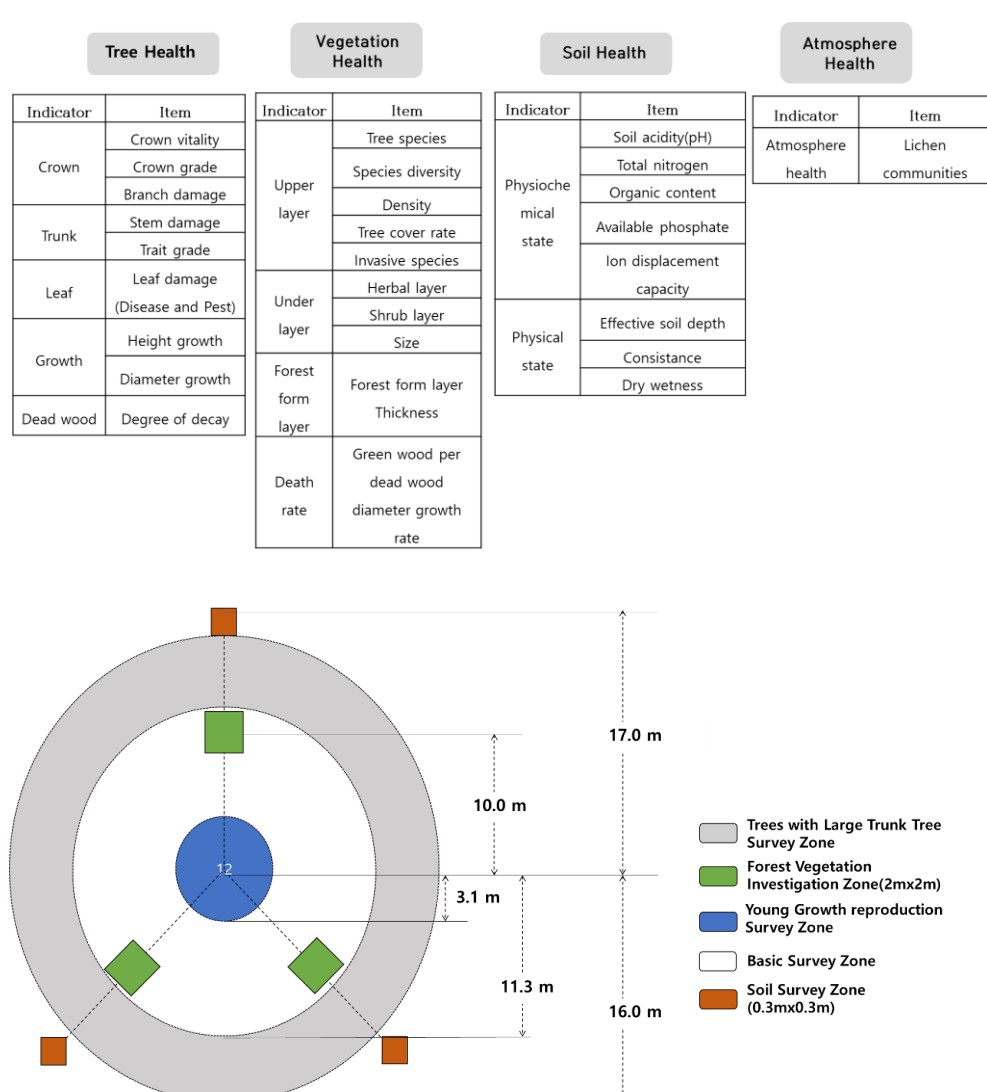

**Figure 2.** The First Forest Health Management Survey Items and Point Structure.

**Table 1.** Target Species.

| Species Name | Scientific Name | Number of Survey Point |
|---|---|---|
| Korean red pine | *Pinus densiflora* | 589 |
| Mongolian oak | *Quercus mongolica* | 445 |
| Konara Oak | *Quercus serrata* | 405 |
| Cork oak | *Quercus variabilis* | 397 |
| Chestnut | *Castanea crenata* | 232 |
| Mountain oriental cherry | *Prunus serrulata* | 208 |
| Sawtooth oak | *Quercus acutissima* | 191 |
| Oriental white oak | *Quercus aliena* | 162 |
| Hua qu liu | *Fraxinus rhynchophylla* | 162 |
| Pitch pine | *Pinus rigida* | 136 |
| Oriental cherry | *Prunus verecunda* | 129 |

## 3. Methods

### 3.1. Species Distribution Modeling

Given the complexity and diversity of plant distribution and the various environmental factors involved, a species distribution model that accounts for multiple factors was used [11]. However, determining the optimal model to represents species distribution

based on sample points can be challenging. To reduce uncertainty that may occur when using a single model and provide consistent and realistic results, an ensemble model was used [12–14]. The ensemble methodology is commonly used in policy decision making to improve distribution prediction and species richness patterns [15]. In this study, we used BIOMOD2, widely recognized as a prominent tool for Ensemble Modeling, in conjunction with the R program to select a model capable of predicting distributions accurately even with a limited number of occurrence points [13]. There are 10 single models that can be run in BIOMOD2. They are primarily categorized into two groups: statistical-based models and machine learning-based models. Statistical-based models include the generalized linear model (GLM), generalized additive model (GAM), and multiple adaptive regression splines (MARS). Machine learning-based models include classification tree analysis (CTA), artificial neural network (ANN), generalized boosting model (GBM), flexible discriminant analysis (FDA), random forest (RF), surface range envelope (SRE), and maximum entropy (MaxEnt) [16].

GLM is a linear regression model, whereas GBM is a machine learning model that aims to approach the actual value, beginning from the mean value, and can be applied to both qualitative and quantitative response variables, similarly to a linear model [17]. In contrast, MARS is a linear regression technique that can model nonlinearity by identifying significant patterns and correlations in the data [18]. FDA, also referred to as linear discriminant analysis, is a classification model that combines linear regression models and uses optimal scores to transform response variables, thus allowing linear separation of data. CTA is a predictive model that connects observations of items and objects using decision trees [19]. RF is a machine learning-based ensemble model that uses decision trees, whereas ANN is a machine learning technique that produces outcomes based on the strength of the connections between variables [20].

In this study, we estimated the species distribution using seven models: GLM, GBM, MARS, FDA, CTA, RF, and ANN. These models were all operated using presence/absence data. GAM is an extension of multiple linear regression; the difference between GAM and GLM, which is a linear regression model, is that the k value representing degrees of freedom is vital, but GAM was excluded because it is ambiguous to set the standard for the k value. We also excluded SRE because continuous numerical analysis was difficult due to the result value not being provided as a continuous numerical value, instead appearing as 0, 0.5, or 1. Furthermore, MaxEnt was excluded because it exhibited vulnerability when the entire range of species was not sampled or when there was variability in the range of species. We deemed MaxEnt inappropriate for the first FHM, which only surveyed forests [21,22]. The modeling process was repeated 30 times for each model, and the ensemble was performed by selecting a value with a receiver operating characteristic (ROC) value of 0.5 or higher [23] (Figure 3).

### 3.2. Environmental Variables

Vegetation distribution is highly sensitive to climatic factors, such as temperature. Therefore, in this study, we incorporated the relevant climate data to investigate this relationship. We obtained data on climatic variables from climate information portal of the Korea Meteorological Administration. The Korea Meteorological Administration provides a raster dataset of approximately 1 km that was generated through a statistical method by applying the MK-PRISM (Modified Korean Parameter-elevation Regressions on Independent Slopes Model) technique. Bioclimatic variables were derived from the MK-PRISM raster dataset using the R Biovar function. These variables were derived from the three climate variables: maximum temperature, minimum temperature, and precipitation. We utilized a total of six variables: Bio1 (annual mean temperature), Bio2 (mean diurnal range), Bio5 (max temperature of warmest month), Bio12 (annual precipitation), Bio13 (precipitation of wettest month), and Bio14 (precipitation of driest month), which were selected based on correlation analysis [13]. We try to avoid multicollinearity by using correlation analysis, we removed Pearson correlation coefficients greater than 0.7 (see Appendix A) [24].

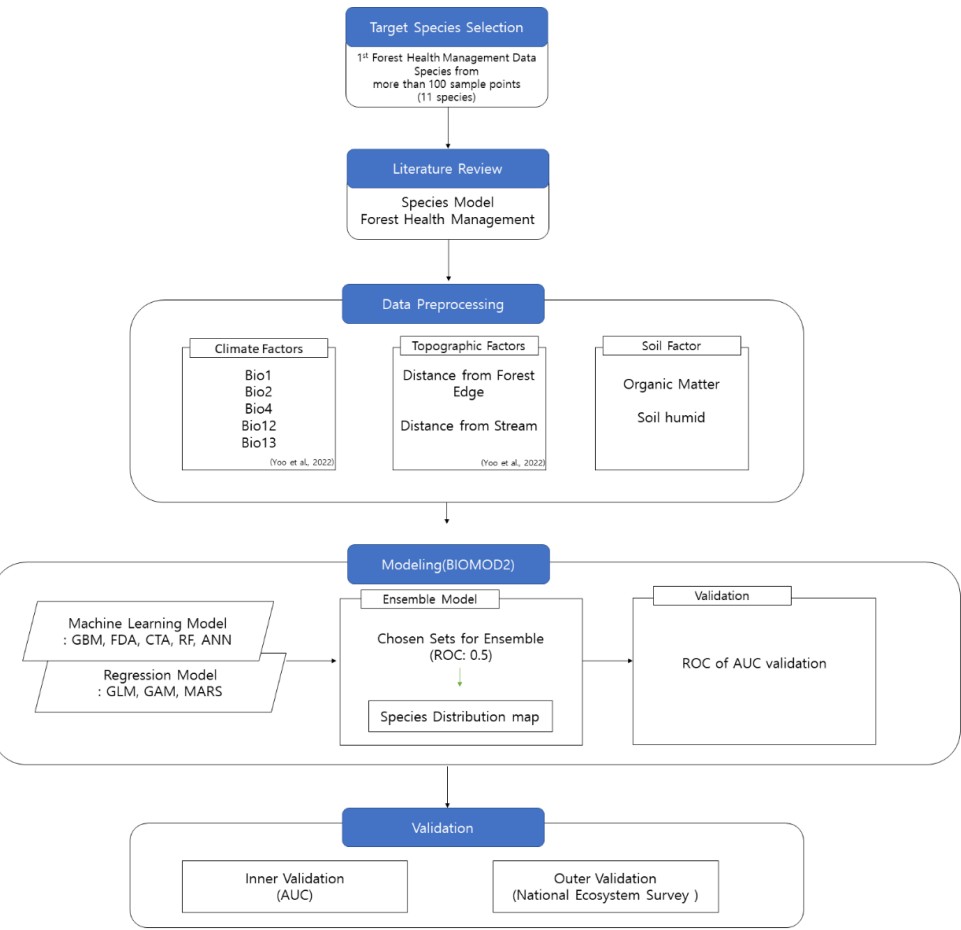

**Figure 3.** Model Operating Flow Chart [13].

To select variables other than climate variables, we referred to previous [9,25] and related studies conducted in Republic of Korea [26–28]. Distance from the forest boundary and distance from the valley were selected as topographical factors, soil and soil organic matter were selected as soil factors, and forest type was selected as the forest factor (Table 2).

**Table 2.** Environmental Variables.

| Category | Variable | Explanation | Reference |
|---|---|---|---|
| Meteorological Factors | Bio1<br>Bio2<br>Bio4<br>Bio12<br>Bio13<br>Bio14 | Annual Mean Temperature<br>Mean Diurnal Range<br>Temperature Seasonality<br>Annual Precipitation<br>Precipitation of Wettest Month<br>Precipitation of Driest Month | Korea Meteorological Administration<br>- Max temperature<br>- Min temperature<br>- Precipitation |
| Soil Factors | Soil humid<br>Soil organic matter | Soil humid<br>Soil organic matter | Korea Forest Service<br>- Forest type map<br>(Soilgrids) |
| Topographic Factors | D_forest | Distance from forest edge | Ministry of Environment<br>- Land cover map |
| | D_valley | Distance from Valley | Water Resources Management Information System<br>- Stream order map |
| Forest Factor | F_type | Forest Type | Korea Forest Service<br>- Forest type map |

*3.3. Accuracy Verification*

3.3.1. Internal Accuracy Verification through Statistical Analysis

The area under the curve (AUC) value of ROC analysis, which is a method used for ensemble verification of the species distribution model, was selected to verify the statistical accuracy of this study [23,29]. ROC is a graph that shows the model performance for all critical values and is mainly used to evaluate the value of AUC. The advantage of using AUC to measure model accuracy is that it is independent of the reference value, hence it is widely used to compare individual models [30–32]. AUC values range from 0.5 to 1, where values of 0.5–0.7 indicate low accuracy, values of 0.7–0.9 are considered normal, and values close to 0.9 or greater than 0.9 indicate an excellent ability to distinguish between the distribution and non-distribution of species. [16,33]. In the development of the species distribution model, the appearance data in the form of points were divided into training and test data (80:20) and repeated five times.

3.3.2. Independent Verification Using External Data

The species richness map created using the ensemble methodology was judged to have limitations in terms of accuracy and statistical verification, and thus required independent verification using other data sources [34,35]; therefore, independent verification was conducted using the National Ecosystem Survey. Since there are no independent data sources for species richness in Korea, the National Natural Environment Survey was selected as it provides data on the appearance of species. The National Ecosystem Survey (NES) (2006–2018) was conducted by the National Institute of Ecology. It began in 1986 and covers the entire Republic of Korea, providing data on the occurrence of endemic, alien, and invasive species. The survey has been conducted every five years since its inception in 1986. This study used the most recent data from the 4th National Natural Environment Survey to conduct independent verification.

To analyze the accuracy of the appearance of each species and compare it with the National Natural Environment Survey, we converted the results of the species distribution model for each species into a binomial distribution of appearance and non-appearance. As the critical point was set differently for each result, we derived the presence/absence map by using a median value of 500 (ranging from 0 to 1000) as the critical point [23].

## 4. Results

*4.1. Species Distribution Model Operating Result*

In this study, we used 7 single models to predict 11 potential habitats in Republic of Korea. To determine the statistical significance of each model, we compared the AUC values of the ROC. The AUC value of the derived ensemble model for each species was higher than 0.8, indicating a high discriminatory ability for presence/pseudo-absence data and that the modeling results were statistically reliable (Figure **??**).

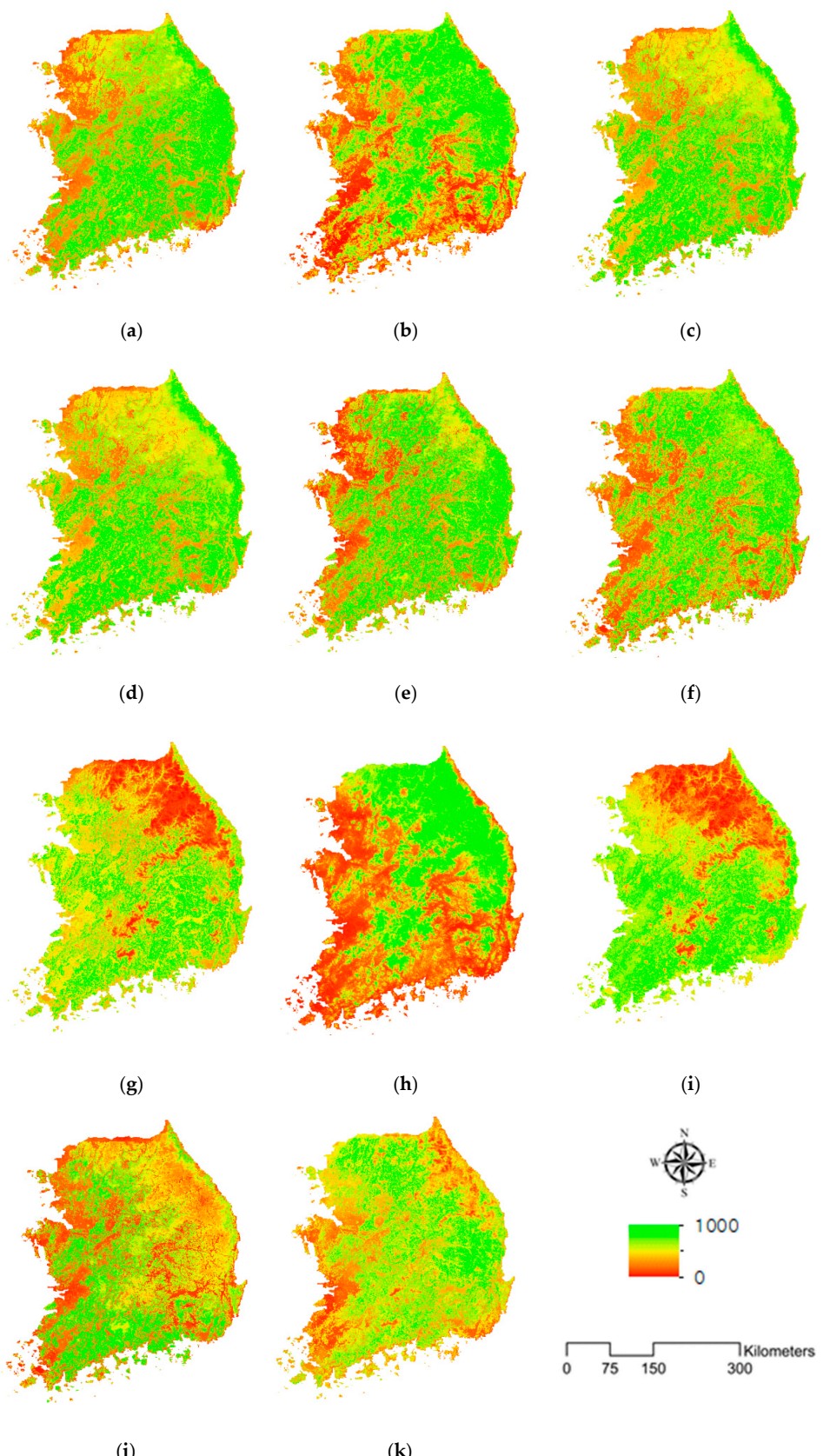

**Figure 4.** Model Operating Result: (**a**) *Pinus Densiflora* (AUC: 0.804), (**b***) Quercus mongolica* (AUC: 0.840), (**c**) *Quercus serrata* (AUC: 0.813), (**d**) *Quercus variabilis* (AUC:0.807), (**e**) *Castanea crenata* (AUC: 0.809), (**f**) *Prunus serrulata* (AUC: 0.804), (**g**) *Quercus acutissima* (AUC: 0.828), (**h**) *Quercus aliena* (AUC: 0.829), (**i**) *Fraxinus rhynchophylla* (AUC: 0.857), (**j**) *Pinus rigida* (AUC: 0.864), (**k**) *Prunus verecunda* (AUC: 0.852).

The contribution of each variable to the prediction of species distribution was determined by analyzing the importance of each variable based on the constructed ensemble model. Since the final result was an ensemble calculated using the simple average method based on the results of each model, the variable importance was also expressed as a simple average (Table 3). The contributions of variables varied for each target species, but in general, the variables with the highest contributions were Bio1 (annual mean temperature) 6 species, organic matter content for 4 species, and distance from forest boundary for one species. Temperature and organic matter content were the most crucial factors for the habitat environment of major tree species in Republic of Korea.

**Table 3.** Variable importance table derived from BIOMOD2 modeling results. Bold values show the highest importance.

| | *C. crenata* | *Q. aliena* | *Q. serrata* | *Q. mongolica* | *P. verecunda* | *Q. variabilis* | *F. rhynchophylla* | *P. densiflora* | *Q. acutissima* | *P. verecunda* | *P. rigida* |
|---|---|---|---|---|---|---|---|---|---|---|---|
| bio01 | 0.0530 | **0.3433** | 0.0947 | **0.3528** | 0.1132 | 0.0583 | **0.5587** | 0.0745 | **0.3453** | **0.3428** | **0.5171** |
| bio02 | 0.0696 | 0.0289 | 0.0418 | 0.0450 | 0.0284 | 0.0713 | 0.0182 | 0.0613 | 0.0300 | 0.0348 | 0.0413 |
| bio04 | 0.0896 | 0.0943 | 0.0781 | 0.0639 | 0.0483 | 0.0872 | 0.0406 | 0.0794 | 0.0918 | 0.1320 | 0.0682 |
| bio12 | 0.1357 | 0.0899 | 0.0424 | 0.0574 | 0.0669 | 0.1318 | 0.0689 | 0.0882 | 0.0850 | 0.1727 | 0.0724 |
| bio13 | 0.1419 | 0.0769 | 0.0767 | 0.0710 | 0.0596 | 0.1528 | 0.0689 | 0.1236 | 0.0882 | 0.1114 | 0.1235 |
| bio14 | 0.0486 | 0.0605 | 0.0653 | 0.0577 | 0.0351 | 0.0545 | 0.0109 | 0.0323 | 0.0578 | 0.1007 | 0.0508 |
| Distance from forest edge | 0.1594 | 0.0875 | 0.0527 | 0.1853 | **0.2200** | 0.1634 | 0.1985 | 0.0602 | 0.0792 | 0.2048 | 0.0255 |
| Soil Organic matter | **0.1938** | 0.0689 | **0.2044** | 0.0529 | 0.0859 | **0.2125** | 0.0220 | **0.3885** | 0.0694 | 0.0647 | 0.0354 |
| Soil Humid | 0.0437 | 0.0885 | 0.0281 | 0.0396 | 0.1019 | 0.0491 | 0.0088 | 0.0248 | 0.0904 | 0.1350 | 0.0944 |
| Forest Type | 0.1765 | 0.1620 | 0.1220 | 0.0978 | 0.2004 | 0.1624 | 0.0097 | 0.0408 | 0.1595 | 0.1149 | 0.0994 |
| Distance From Valley | 0.1123 | 0.0879 | 0.0345 | 0.0475 | 0.0784 | 0.1085 | 0.0461 | 0.0415 | 0.0918 | 0.0211 | 0.0458 |

### 4.2. Binomial Data Result

To verify the prediction accuracy of the species distribution model, the model was converted to binomial data based on the median value. The resulting map was then reclassified as binomial data and overlapped with NES data, which allowed us to confirm whether the model accurately predicted the actual distribution of species. The confirmation of tree species distribution showed an accuracy rate of more than 70% in most species, except for *Quercus mongolica*, which showed the accuracy rate below 70%. This discrepancy is due to the differences in the distribution of *Quercus mongolica* in urban forests in NES, which surveyed all terrains across the country, unlike the data from the first FHM, which only surveyed mountain forests (Table 4).

**Table 4.** Binomial Data Accuracy Rate.

| Species Name | Scientific Name | Accuracy Percent |
|---|---|---|
| Korean red pine | *Pinus densiflora* | 77% |
| Mongolian oak | *Quercus mongolica* | 76% |
| Konara Oak | *Quercus serrata* | 79% |
| Cork oak | *Quercus variabilis* | 79% |
| Chestnut | *Castanea crenata* | 70% |
| Mountain oriental cherry | *Prunus serrulata* | 79% |
| Sawtooth oak | *Quercus acutissima* | 74% |
| Oriental white oak | *Quercus aliena* | 66% |
| Hua qu liu | *Fraxinus rhynchophylla* | 73% |
| Pitch pine | *Pinus rigida* | 74% |
| Oriental cherry | *Prunus verecunda* | 70% |

*4.3. Production and Utilization of Species Richness Map*

A map was created to represent the richness of 11 tree species, indicated by a scale of 0–11. This was accomplished by overlapping the distribution maps of the 11 species into a binomial map (Figure 5a). To validate the accuracy of the map, the vegetation climate distribution map prepared by the Korea National Arboretum was used (Figure 5b). We concluded that species richness was high around the boundary between the northern temperate region and deciduous broad-leaved forest in the central temperate region. This was due to the distribution of three deciduous broad-leaved tree species, *Quercus mongolica*, *Quercus serrata*, and *Quercus aliena*, as well as *Pinus densiflora*, a representative evergreen coniferous species, and the results of the *Pinus rigida* afforestation policy in the Chungcheong-do region [36], including the temperate and southern regions. Furthermore, the distribution of *Fraxinus rhynchophylla* in the temperate northern region appears to have contributed to the observed high species richness.

Based on the analysis of the land cover map, we found low values primarily in urban areas and farmlands, where forests are absent. As species richness was derived mainly in the urban boundary area, it seems that the results of the second urban forest master plan are visible [37], a forest policy in which forest management and monitoring were actively carried out mainly in urban forests in the urban boundary area. The high value of species richness mainly focused on the urban boundary area, which aligns with the implementation of the second urban forest master plan (Figure 6).

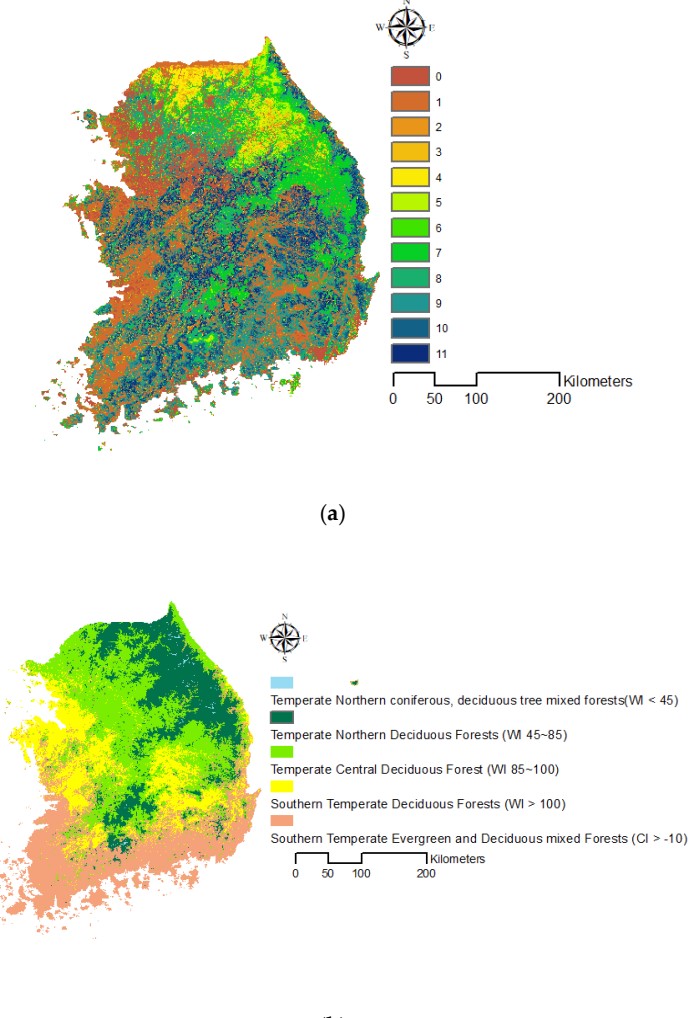

**Figure 5.** Comparison of species richness map and vegetation climate map: (**a**) Species Richness Map, (**b**) Vegetation Climate Distribution Map (WI: Warmth Index, CI: Coldness Index).

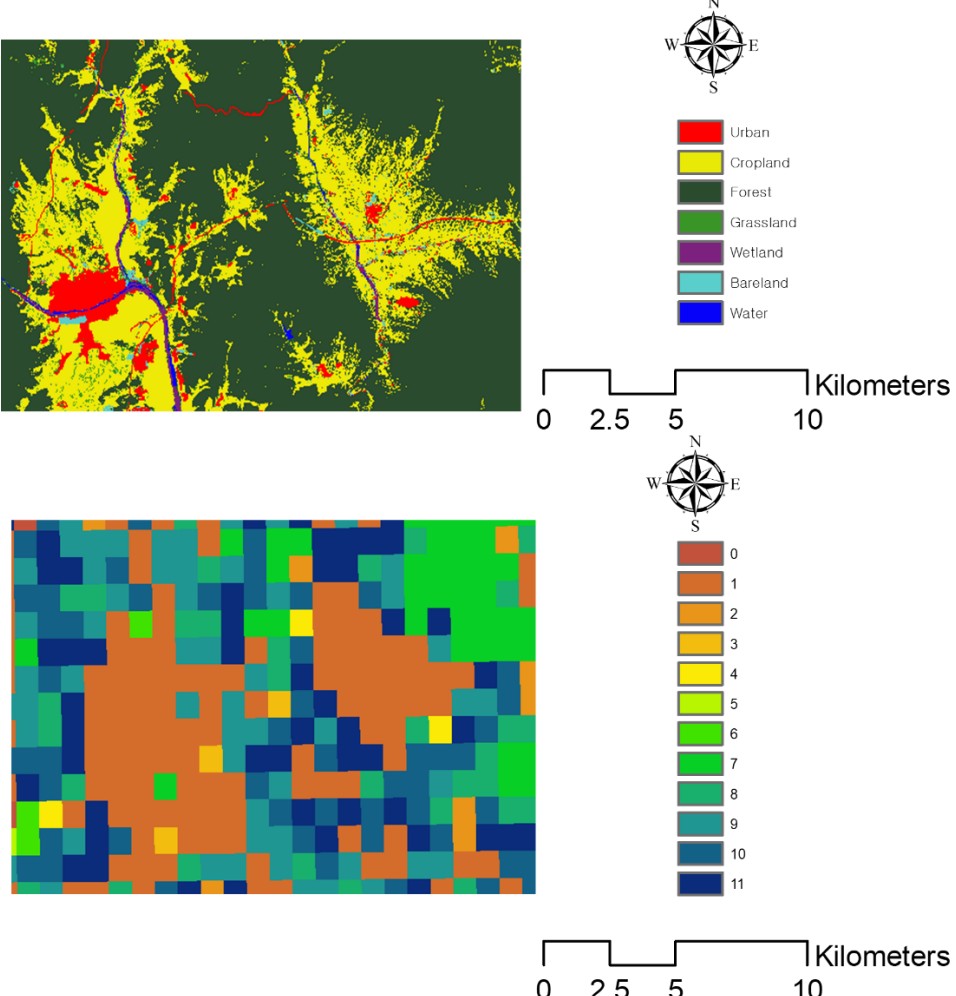

**Figure 6.** Comparative example of species richness map and land cover map.

## 5. Discussions

In this study, we conducted an ensemble modeling of species distribution for eleven woody plant species that are major habitats in Republic of Korea. Based on the ensemble modeling result, it was converted into a binomial map and then overlapped to create the species richness map. The statistical verification using BIOMOD revealed an AUC value as high as 0.8, and independent verification using the NES confirmed that all but one species had an accuracy rate of 70% or higher. These results demonstrate the high accuracy of the modeling approach, indicating that the results are valid. Based on the results, the species richness map created using this methodology was judged to be suitable for use as spatial data for forest management.

As a result of variable contribution analysis, the results were different for each target species, but the Bio1 (annual average temperature) variable appeared the highest, followed by soil organic matter.

The warming trend announced by the IPCC is 0.1 °C/10 years, whereas the warming trend in the Korean Peninsula is 0.15 °C/10 years, indicating that the warming trend in the Korean Peninsula is steeper [38]. Since the tree species with the highest average annual temperature showed a high contribution, it was confirmed that climate change is an essential factor to be considered for forest management in the future.

Soil organic matter content is highly related to precipitation, and as precipitation increases, there is a risk of soil organic matter being lowered [39]. Because Korea will change to a subtropical climate in the future and have a characteristic of increasing annual

precipitation [40], a forest policy based on climate change will be needed in terms of forest soil organic matter that affects tree species richness.

To find a way to utilize the tree species richness map, the distribution was compared using the vegetation distribution climate map and the land cover map as comparative data. The vegetation climate distribution map was selected as comparative data to link climate change policy in the future, and the land cover map was selected to confirm how land cover characteristics affected species richness.

Based on the analysis of the vegetation climate distribution map, evergreen broad-leaved mixed forests in the temperate southern region were derived at a low level. We found that southern temperate evergreen and deciduous mixed forests zone had low species richness, which might be due to the fact that these forests belong to a warm climate zone. Because the first FHM survey did not include warm zone plants in its target species, low species richness was observed. Therefore, it is suggested that it is necessary to check the distribution of temperate plants in this region when implementing forest policy.

Based on the analysis of the land cover map, we found low values primarily in urban areas and farmlands. Forests in the urban center or farmland in Republic of Korea were found to have low species richness, suggesting that the area requires closer investigation and management.

## 6. Conclusions

This study investigated a method that can contribute to the species richness of forests in Republic of Korea by estimating the species richness of trees in Republic of Korean forests where field survey data are unavailable. To achieve this, we used data from the first FHM survey, which is a field survey that provides information on species diversity or species richness at the sampling point level, but with no spatial reference. Due to the complex topography and different forest types in Republic of Korea, model prediction using only one indicator is difficult because no verification data are available. Unlike previous studies, which could only evaluate information within a sample point, this study is meaningful in that it visually extends to a spatial range.

This study employed the species distribution ensemble model to model species richness in Republic of Korean forests. Due to the complex topography of the region and varying forest characteristics, it was difficult to spatially represent the species richness of forests accurately. Therefore, a modeling method was applied to take this into account. Eleven species commonly found in Republic of Korea were selected, following the example of a forest health analysis conducted in the UK [7]. A species richness map of Republic of Korean forests was created by applying a species distribution ensemble model (BIOMOD) to the selected species.

The results of this study have significant implications for the prediction of species richness in areas where survey data are unavailable, using species survey data from the first FHM dataset. This dataset is particularly noteworthy for its diverse range of survey items among existing forest resource survey data. Because data from the first FHM survey were used, we believe that there is a future opportunity to analyze the health of Republic of Korean forests through linkage with other survey indicators.

**Author Contributions:** Conceptualization, J.L. and Y.Y.; Data curation, J.L.; Investigation, J.L.; Methodology, J.L. and Y.Y.; Project administration, R.J.; Supervision, S.J. Validation: R.J.; Writing—original draft, J.L., Y.Y. and R.J.; Writing—review and editing, S.J. All authors have read and agreed to the published version of the manuscript.

**Funding:** This work was supported by the Korea Environment Industry and Technology Institute (KEITI) through the Decision Support System Development Project for Environmental Impact Assessment, funded by the Korea Ministry of Environment (MOE) (No. 2020002990009).

**Data Availability Statement:** Not applicable.

**Conflicts of Interest:** The authors declare no conflict of interest.

## Appendix A

**Table A1.** Pearson's correlation coefficient matrix of bioclimatic variables. The values which have a correlation ($p > 0.7$) have been changed into red.

| 1 | Bio1 | Bio2 | Bio3 | Bio4 | Bio5 | Bio6 | Bio7 | Bio8 | Bio9 | Bio10 | Bio11 | Bio12 | Bio13 | Bio14 | Bio15 | Bio16 | Bio17 | Bio18 | Bio19 |
|---|---|---|---|---|---|---|---|---|---|---|---|---|---|---|---|---|---|---|---|
| Bio1 | 1.000 | | | | | | | | | | | | | | | | | | |
| Bio2 | −0.510 | 1.000 | | | | | | | | | | | | | | | | | |
| Bio3 | −0.251 | 0.879 | 1.000 | | | | | | | | | | | | | | | | |
| Bio4 | −0.599 | 0.451 | −0.015 | 1.000 | | | | | | | | | | | | | | | |
| Bio5 | 0.821 | 0.118 | 0.062 | 0.156 | 1.000 | | | | | | | | | | | | | | |
| Bio6 | 0.878 | −0.708 | −0.334 | −0.838 | 0.224 | 1.000 | | | | | | | | | | | | | |
| Bio7 | −0.656 | 0.762 | 0.362 | 0.909 | 0.150 | −0.930 | 1.000 | | | | | | | | | | | | |
| Bio8 | 0.824 | −0.333 | −0.315 | −0.083 | 0.867 | 0.539 | −0.220 | 1.000 | | | | | | | | | | | |
| Bio9 | 0.888 | −0.472 | −0.106 | −0.830 | 0.329 | 0.899 | −0.788 | 0.550 | 1.000 | | | | | | | | | | |
| Bio10 | 0.768 | −0.215 | −0.250 | 0.043 | 0.928 | 0.422 | −0.077 | 0.963 | 0.449 | 1.000 | | | | | | | | | |
| Bio11 | 0.917 | −0.525 | −0.129 | −0.864 | 0.325 | 0.962 | −0.853 | 0.561 | 0.963 | 0.462 | 1.000 | | | | | | | | |
| Bio12 | −0.073 | 0.104 | 0.291 | −0.298 | −0.255 | 0.082 | −0.180 | −0.245 | 0.210 | −0.328 | 0.115 | 1.000 | | | | | | | |
| Bio13 | −0.395 | 0.101 | −0.100 | 0.432 | −0.164 | −0.408 | 0.352 | −0.193 | −0.361 | −0.190 | −0.460 | 0.609 | 1.000 | | | | | | |
| Bio14 | 0.041 | −0.135 | 0.091 | −0.387 | −0.247 | 0.287 | −0.385 | −0.140 | 0.206 | −0.231 | 0.240 | 0.526 | 0.076 | 1.000 | | | | | |
| Bio15 | −0.432 | 0.136 | −0.292 | 0.848 | 0.082 | −0.636 | 0.676 | −0.018 | −0.673 | 0.087 | −0.703 | −0.220 | 0.608 | −0.491 | 1.000 | | | | |
| Bio16 | −0.293 | 0.205 | 0.213 | 0.067 | −0.252 | −0.222 | 0.130 | −0.307 | −0.112 | −0.331 | −0.210 | 0.898 | 0.851 | 0.344 | 0.208 | 1.000 | | | |
| Bio17 | 0.137 | −0.128 | 0.176 | −0.533 | −0.219 | 0.396 | −0.484 | −0.110 | 0.365 | −0.226 | 0.378 | 0.710 | 0.124 | 0.931 | −0.591 | 0.466 | 1.000 | | |
| Bio18 | −0.299 | 0.200 | 0.184 | 0.123 | −0.208 | −0.233 | 0.158 | −0.251 | −0.136 | −0.289 | −0.237 | 0.888 | 0.875 | 0.334 | 0.229 | 0.978 | 0.458 | 1.000 | |
| Bio19 | 0.152 | −0.129 | 0.177 | −0.544 | −0.210 | 0.403 | −0.489 | −0.099 | 0.390 | −0.217 | 0.392 | 0.726 | 0.130 | 0.924 | −0.597 | 0.477 | 0.998 | 0.468 | 1.000 |

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
