# Peer review of "Mapping the Species Richness of Woody Plants in Republic of Korea"

_sustainability, doi:10.3390/su15075718_

Round 1

Reviewer 1 Report

2.2. plots of fixed sample points were surveys

Use dot instead of comma in numbers in text and tables

species scientific names throughouth the text need to be reviewed.

Author Response

RESPONSES TO REVIEWER’S COMMENTS

Journal: MDPI Sustainability

Manuscript ID: sustainability-2266550

Title: Mapping the species richness of woody plants in South Korea

Authors: Junhee Lee, Youngjae Yoo, Raeik Jang and Seong-woo Jeon

We would like to express our appreciation for your detailed and helpful comments on our manuscript. Your advice was very helpful in improving our manuscript by adding necessary explanations of the rationale for the study design and the significance of the simulation results so that we can better communicate with our readers. Based on your comments, we have revised the manuscript thoroughly while attempting to incorporate your comments into the article as much as we can. If you note any further shortcomings, we welcome your comments. We hope the revised manuscript will meet the standards of publication. Details of our revisions are provided in the following sections.

Reviewer 1’s Comments and Suggestions:

This paper presents an application of Mapping the species richness of woody plants in South Korea. It is a topic of interest to the researchers in the related areas, but the application proposed is not novel and an explanation of why the authors did these various simulations is not provided. Furthermore, there are few explanations of the rationale for the study design. Therefore, I suggest the author did the major revision before publication. The detailed comments are as follows:

Major Comments

Answers

1

2.2. plots of fixed sample points were surveys

Use dot instead of comma in numbers in text and tables

.

We appreciate your comment. Dots instead of commas can confuse numbers, so commas are used. Commas are used instead of dots to represent the number four thousand(4,000).

2

species scientific names throughouth the text need to be reviewed

Thanks for your comment. We checked again Scientific name.

Reviewer 2 Report

The paper "Mapping the species richness of woody plants in South Korea" is an interesting topic and fits to the journal scope. However, paper was written poorly and not sure if the author or authors know how to write peer-review article in standard journal (usually abstract, introduction, methods and materials, results, discussion, conclusions and references sections present). It looks like written by undergraduate students. There is organization problem, no discussion section included or discussed the results, conclusion section is mess and references are not provided in many places. In current shape, the paper should not be published. I'd encourage authors to rewrite the paper, make it simple and clear, add all the required standard sections in the paper and resubmit again. Some of results are very interested and well-presented. Overall, introduction, methods and results sections are not bad. But require to work on other sections. Also, research questions are not clear, neither the expectations out of the paper.

Other minor comments are provided below.

Lines 75-84: Not clear the research question (s). Nor provided relevant references in many places, e.g., "Based on our findings, we confirmed the distribution of species richness through a comparison with the vegetation climate distribution map, which is used in climate change adaptation policy."- reference needed. Also, what is your finding? You have not p resented your results yet. Remove this sentence or say this study will be beneficial or useful for.....

Line 100- What are these 28 items and 12 survey indicators? How the readers know this? Clear it up.

Line 105-106: Again 28 items? What are these?

107-108: Where is the reference to say this?

110-112: I don't see relevance here, your study in Korea but based on most of abundant species in the UK? I'm confused. Make it clear.

158-160: Provide reference that you set the criteria of ROC.

173-176: You can't say based on previous study, you have to show multicollinearity results among all the variables and select the one that not highly correlated (usually in literature used, r<0.7), look at the Guisan et al. 2017 (Habitat Suitability and Distribution Models) reference.  Present the multicollinearity results for the selection (may be as supplementary file).

177-179: Even if you used variables based on previous study you need to justify why you think these are the important variables for your study.

180: What is forestry? explain it.

There is no discussion section or results are not discussed. Not sure what are these results means. Needs to add discussion section or can include as Results and Discussion one section (Very important).

Conclusions  and Implications section does not make sense. I am not sure what is concluded. Need to be clear and to the point based on results. Don;t talk too many limitations and errors, that means your research is not good. Rewrite the whole section and conclude based on your results.

Exclude implications and limitations from conclusion. Move them the discussion section (you need it) and add a section implications and limitations where you can put it whole it.

Author Response

RESPONSES TO REVIEWER’S COMMENTS

Journal: MDPI Sustainability

Manuscript ID: sustainability-2266550

Title: Mapping the species richness of woody plants in South Korea

Authors: Junhee Lee, Youngjae Yoo, Raeik Jang and Seong-woo Jeon

We would like to express our appreciation for your detailed and helpful comments on our manuscript. Your advice was very helpful in improving our manuscript by adding necessary explanations of the rationale for the study design and the significance of the simulation results so that we can better communicate with our readers. Based on your comments, we have revised the manuscript thoroughly while attempting to incorporate your comments into the article as much as we can. If you note any further shortcomings, we welcome your comments. We hope the revised manuscript will meet the standards of publication. Details of our revisions are provided in the following sections.

Reviewer 2’s Comments and Suggestions:

This paper presents an application of Mapping the species richness of woody plants in South Korea. It is a topic of interest to the researchers in the related areas, but the application proposed is not novel and an explanation of why the authors did these various simulations is not provided. Furthermore, there are few explanations of the rationale for the study design. Therefore, I suggest the author did the major revision before publication. The detailed comments are as follows:

Major Comments

Answers

1

Lines 75-84: Not clear the research question (s). Nor provided relevant references in many places, e.g., "Based on our findings, we confirmed the distribution of species richness through a comparison with the vegetation climate distribution map, which is used in climate change adaptation policy."- reference needed. Also, what is your finding? You have not p resented your results yet. Remove this sentence or say this study will be beneficial or useful for.....

After receiving your comment, we realized that our original manuscript lacked sufficient explanation. The sentence you pointed out has been deleted. In addition to that, we will add additional explanations to the discussion section.

2

Line 100- What are these 28 items and 12 survey indicators? How the readers know this? Clear it up.

Line 105-106: Again 28 items? What are these?

Thank you for your comment. We included the figure for understanding of 28 items of Forest Health Management Data.

Page 3 , Lines 111

3

107-108: Where is the reference to say this?

We appreciate that your comment. It helped us improve our manuscript. Number 10 of the Reference part explains this. We added it

Page 3, Lines 104-105.

In this study, the species diversity index, which can best express vegetation health among the seven indicators, was calculated based on species composition data[10].

4

110-112: I don't see relevance here, your study in Korea but based on most of abundant species in the UK? I'm confused. Make it clear.

Thank you for your comment. When researching forest health in the UK, we referred to the case of conducting a study by selecting the most common species. There were 5 species selected in the UK, 11 species in this study. As mentioned, the word 5 species can be confusing, so we excluded five. A study was conducted targeting the most common 11 species in South Korea.

Page 3, Lines 107-111.

We selected the most abundant species in Korea as its target species by referring to a previous study [7] that had evaluated forest health for the most abundant species in the UK.

5

158-160: Provide reference that you set the criteria of ROC.

Thank you for your comment. We added the reference. Number 28 of the Reference part explains this. We added it

Page 5, Lines 157-158.

 The modeling process was repeated 30 times for each model, and the ensemble was performed by selecting a value with a receiver operating characteristic (ROC) value of 0.5 or higher[28].

6

173-176: You can't say based on previous study, you have to show multicollinearity results among all the variables and select the one that not highly correlated (usually in literature used, r<0.7), look at the Guisan et al. 2017 (Habitat Suitability and Distribution Models) reference.  Present the multicollinearity results for the selection (may be as supplementary file).

We appreciate your comment. After reading your comment, we agreed that the reference was inappropriate. As you suggested, the process of deducing variables has been added to the Appendix A. We will add a table containing Pearson's correlation coefficient for each bioclimatic variable. Thank you for helping us improve our manuscript.

Page 6, Lines 174-176:

We try to avoid multicollinearity by using correlation analysis, we removed Pearson correlation coefficient greater than 0.7.

7

180: What is forestry? explain it.

We appreciate your comment, but we decided to delete forestry in the manuscript.  As mentioned, the word Forestry can cause confusion, so it has been modified to Forest type.

8

There is no discussion section or results are not discussed. Not sure what are these results means. Needs to add discussion section or can include as Results and Discussion one section (Very important).

Your comment is greatly appreciated. After reading your comments, we tried to revise the manuscript thoroughly with a better organizational focus on the Discussion section. The purpose of this study is to construct a species richness map and propose to use it for forest management policy and climate change policy. I wrote a discussion part focusing on this point. As you said, the part about research limitations was also reduced. All the revisions made to the manuscript have been colored blue for easier identification. I truly hope you find our revision to meet the standards of publication.

Page 12, Lines 276- 316:

5. Discussions

In this study, we conducted an ensemble modeling of species distribution for eleven woody plant species that are major habitats in South Korea. Based on ensemble modeling result, it was converted into a binomial map and then overlapped to create the species richness map. The statistical verification using BIOMOD revealed an AUC value as high as 0.8, and independent verification using the NES confirmed that all but one species had an accuracy rate of 70% or higher. These results demonstrate the high accuracy of the modelling approach, indicating that the results are valid. Based on the results, species richness map created using this methodology was judged to be suitable for use as spatial data for forest management.

As a result of variable contribution analysis, the results were different for each target species, but Bio1 (annual average temperature) variable appeared the highest, followed by soil organic matter.

The warming trend announced by the IPCC is 0.1℃/10 years, whereas the warming trend in the Korean Peninsula is 0.15℃/10 years, indicating that the warming trend in the Korean Peninsula is steeper [37]. Since the tree species with the highest average annual temperature showed a high contribution, it was confirmed that climate change is an essential factor to be considered for forest management in the future.

Soil organic matter content is highly related to precipitation, and as precipitation increases, there is a risk of soil organic matter being lowered [38]. Since Korea will change to a subtropical climate in the future and have a characteristic of increasing annual precipitation[39], a forest policy based on climate change will be needed in terms of forest soil organic matter that affects tree species richness.

To find a way to utilize the tree species richness map, the distribution was compared using the vegetation distribution climate map and the land cover map as comparative data. The vegetation climate distribution map was selected as comparative data to link climate change policy in the future, and the land cover map was selected to confirm how land cover characteristics affected species richness.

 Based on the analysis of the vegetation climate distribution map, evergreen broad-leaved mixed forests in the temperate southern region were derived at a low level. We found that southern temperate evergreen and deciduous mixed forests zone had low species richness, which might be due to the fact that these forests belong to a warm climate zone. Since the first FHM survey did not include warm zone plants in its target species, low species richness observed. Therefore, it was derived that it is necessary to check the distribution of temperate plants in this region when implementing forest policy.

 Based on the analysis of the land cover map, we found low values primarily in urban areas and farmlands Forests in the urban center or farm land in the south korea were found to have low species richness, suggesting that the area requires closer investigation and management.

Appendix A

Table A1. Pearson’s correlation coefficient matrix of bioclimatic variables. The value which has correlated(p > 0.7) had been changed into red

Bio1

Bio2

Bio3

Bio4

Bio5

Bio6

Bio7

Bio8

Bio9

Bio10

Bio11

Bio12

Bio13

Bio14

Bio15

Bio16

Bio17

Bio18

Bio19

Bio1

1.000

Bio2

-0.510

1.000

Bio3

-0.251

0.879

1.000

Bio4

-0.599

0.451

-0.015

1.000

Bio5

0.821

0.118

0.062

0.156

1.000

Bio6

0.878

-0.708

-0.334

-0.838

0.224

1.000

Bio7

-0.656

0.762

0.362

0.909

0.150

-0.930

1.000

Bio8

0.824

-0.333

-0.315

-0.083

0.867

0.539

-0.220

1.000

Bio9

0.888

-0.472

-0.106

-0.830

0.329

0.899

-0.788

0.550

1.000

Bio10

0.768

-0.215

-0.250

0.043

0.928

0.422

-0.077

0.963

0.449

1.000

Bio11

0.917

-0.525

-0.129

-0.864

0.325

0.962

-0.853

0.561

0.963

0.462

1.000

Bio12

-0.073

0.104

0.291

-0.298

-0.255

0.082

-0.180

-0.245

0.210

-0.328

0.115

1.000

Bio13

-0.395

0.101

-0.100

0.432

-0.164

-0.408

0.352

-0.193

-0.361

-0.190

-0.460

0.609

1.000

Bio14

0.041

-0.135

0.091

-0.387

-0.247

0.287

-0.385

-0.140

0.206

-0.231

0.240

0.526

0.076

1.000

Bio15

-0.432

0.136

-0.292

0.848

0.082

-0.636

0.676

-0.018

-0.673

0.087

-0.703

-0.220

0.608

-0.491

1.000

Bio16

-0.293

0.205

0.213

0.067

-0.252

-0.222

0.130

-0.307

-0.112

-0.331

-0.210

0.898

0.851

0.344

0.208

1.000

Bio17

0.137

-0.128

0.176

-0.533

-0.219

0.396

-0.484

-0.110

0.365

-0.226

0.378

0.710

0.124

0.931

-0.591

0.466

1.000

Bio18

-0.299

0.200

0.184

0.123

-0.208

-0.233

0.158

-0.251

-0.136

-0.289

-0.237

0.888

0.875

0.334

0.229

0.978

0.458

1.000

Bio19

0.152

-0.129

0.177

-0.544

-0.210

0.403

-0.489

-0.099

0.390

-0.217

0.392

0.726

0.130

0.924

-0.597

0.477

0.998

0.468

1.000

Reviewer 3 Report

Dear authors, I liked the article. The article provides an analysis of the conducted modeling on the distribution of woody plant species, which are the main habitats in South Korea. A species richness map was created to promote tree species diversity to assess the health of forests in South Korea. It quite fully reflects and characterizes the range of conceived ideas.
The topic chosen by the authors is relevant and original. Since on the Korean peninsula, especially its southern part, there is a very strong influence of climatic and anthropogenic factors on the environment, including forests. This is especially noticeable in species richness.
An independent verification was carried out using statistical methods. The test took into account the adaptation of forests to climate change. The data obtained confirmed that woody plant species richness is highest around the border of deciduous forests in the central temperate zone and lowest around evergreen and deciduous mixed forests in the southern temperate zone of the Korean Peninsula.
The authors have chosen the right path of methodology. All conclusions are presented by evidence and reasoned. The findings answer the questions posed by this study.  
I have no additional comments on the figures and tables. In my opinion they are made at a good level and readable. All reference links are relevant in my opinion.
Of course, any scientific work may have questions and claims. But I have no fundamental objections to this manuscript, let alone claims. The work is good, well done, and most importantly very clear interface and logic!

Author Response

RESPONSES TO REVIEWER’S COMMENTS

Journal: MDPI Sustainability

Manuscript ID: sustainability-2266550

Title: Mapping the species richness of woody plants in South Korea

Authors: Junhee Lee, Youngjae Yoo, Raeik Jang and Seong-woo Jeon

We would like to express our appreciation for your detailed and helpful comments on our manuscript. Your advice was very helpful in improving our manuscript by adding necessary explanations of the rationale for the study design and the significance of the simulation results so that we can better communicate with our readers. Based on your comments, we have revised the manuscript thoroughly while attempting to incorporate your comments into the article as much as we can. If you note any further shortcomings, we welcome your comments.

Round 2

Reviewer 2 Report

Authors did a decent job. I am satisfied with the revision.